# Interhelical H-Bonds Modulate the Activity of a Polytopic Transmembrane Kinase

**DOI:** 10.3390/biom11070938

**Published:** 2021-06-25

**Authors:** Juan Cruz Almada, Ana Bortolotti, Jean Marie Ruysschaert, Diego de Mendoza, María Eugenia Inda, Larisa Estefanía Cybulski

**Affiliations:** 1Departamento de Microbiología, Facultad de Ciencias Bioquímicas y Farmacéuticas, Universidad Nacional de Rosario-Argentine National Research Council-CONICET, Suipacha 531, Rosario 2000, Argentina; almadajuancruz@gmail.com (J.C.A.); anabortolotti@gmail.com (A.B.); 2Laboratory for the Structure and Function of Biological Membranes, Center for Structural Biology and Bioinformatics, Université Libre de Bruxelles, CP 206/2, Bd du Triomphe, 1050 Brussels, Belgium; 3Laboratorio de Fisiología Microbiana, Instituto de Biología Molecular y Celular de Rosario (IBR), CONICET, Facultad de Ciencias Bioquímicas y Farmacéuticas, Universidad Nacional de Rosario, Ocampo y Esmeralda, Predio CONICET Rosario, Rosario 2000, Argentina; demendoza@ibr-conicet.gov.ar; 4Research Laboratory of Electronics, Department of Electrical Engineering and Computer Science, Massachusetts Institute of Technology, Cambridge, MA 02139, USA; 5MIT Synthetic Biology Center, Massachusetts Institute of Technology, Cambridge, MA 02139, USA

**Keywords:** transmembrane protein interactions, hydrogen bond interaction, signal transduction, histidine kinase, dimerisation motif, receptor

## Abstract

DesK is a Histidine Kinase that allows *Bacillus subtilis* to maintain lipid homeostasis in response to changes in the environment. It is located in the membrane, and has five transmembrane helices and a cytoplasmic catalytic domain. The transmembrane region triggers the phosphorylation of the catalytic domain as soon as the membrane lipids rigidify. In this research, we study how transmembrane inter-helical interactions contribute to signal transmission; we designed a co-expression system that allows studying in vivo interactions between transmembrane helices. By Alanine-replacements, we identified a group of polar uncharged residues, whose side chains contain hydrogen-bond donors or acceptors, which are required for the interaction with other DesK transmembrane helices; a particular array of H-bond- residues plays a key role in signaling, transmitting information detected at the membrane level into the cell to finally trigger an adaptive response.

## 1. Introduction

Histidine kinases (HK) are the sensory proteins of two-component systems. They perceive and respond to a broad range of stimuli that may imply general stress or specific signals, such as the concentration of particular nutrients or ions. Upon signal detection, HKs phosphorylate their cognate response regulator in the cytoplasm.

Almost all of the prototypical HK characterised structurally are homo-dimeric, and consist of three modules [1,2]. The extracellular region acts as a sensor, detecting changes in the surrounding medium, and the transmembrane (TM) region transmits the signal from the extracellular to the intracellular compartment. Finally, the cytoplasmic region is a well-conserved kinase domain that has two sub-domains: the catalytic and ATP-binding domain (CA) and the Dimerisation and Histidine phosphotransfer domain (DHp), which phosphorylates its cognate response regulator. The HK controls the phosphorylation state of the RR, thereby modulating the RR DNA-binding capacity [1,2].

Here we study *Bacillus subtilis* DesK, a membrane-bound HK part of a two-component system that maintains appropriate membrane fluidity by regulating the levels of unsaturated fatty acids (Figure 1A, left panel). Changes in membrane lipid fluidity that occur as a consequence of different environmental conditions (such as nutrients, temperature or pH) modulate the activity of the sensor [3,4,5]. DesK lacks the extracellular sensor domain; instead, sensing is carried out by 10 transmembrane helices (coming from two monomers) arranged in an unknown structure. Interestingly, a DesK construct lacking the TM domain has constitutive activity, suggesting that the sensor domain locates in the TM region [6].

Genetic and biochemical evidence suggests that DesK, a Class I HK, exists in two alternative conformational states, kinase and phosphatase, and the presence of the stimulus favours the transition between these states [7,8].

DesK is inactive when the membrane is fluid and thin, as it occurs at 37 °C, and turns activated when the membrane is more rigid and thick, as it occurs when temperature drops below 25 °C [4,9]. It has a cytoplasmic catalytic domain DesKC, including subdomains CA and DHp, as revealed in several X-ray structures [7,10] and five transmembrane segments (TMSs), which are required for sensing membrane properties [9]. When activated, a DesK dimer undergoes auto-phosphorylation and the phosphate group is then transferred to the response regulator, DesR. Phosphorylated DesR binds as a tetramer to the promoter of the gene encoding a Desaturase, and activates its transcription [11]. This Desaturase is a transmembrane enzyme, which introduces double bonds in the fatty acids of the lipids, making the membrane more fluid (Figure 1A) [4,5].

To study the sensing mechanism, which implies 10 TMS, we designed and built a simplified version of DesK, which was called Minimal Sensor-DesKC (MS-DesKC). This is an engineered protein, in which the cytoplasmic domain is fused to a single membrane-spanning alpha helix that contains the necessary and sufficient components required for signaling [9]. The merged functional TMS was built by fusing 17 residues corresponding to the N-terminus of TM1 with 14 residues of the C-terminus of TM5 and the cytoplasmic catalytic domain of DesK (Figure 1A, right panel). This hybrid construction mimics the full-length protein: it is more active at 25 °C than at 37 °C. As the MS-DesKC lacks an important portion of the TM domain, its regulation is not as tight as that of the full length protein; nevertheless, it allowed us to study at molecular level the mechanism of thermosensing [12,13]. We have demonstrated that a group of Serine residues located towards the C-terminus of TM5 could work as a zipper to transmit the signal [14]. In fluid membranes, which are more hydrated, the high dielectric constant hampers strong interactions between inter-helical hydroxyl groups of Serine residues. In rigid and dehydrated membranes, the hydroxyl groups of Serine (located in different helices) interact more strongly, closing the zipper and transmitting the signal [14]. This suggests that formation of interhelical H-bonds leads to changes in the transmembrane dimerisation interface, and this could be directly related to a particular signaling state. Most of the experiments that have built this model were performed mainly with the single-pass hybrid MS-DesKC. Some experimental studies of the five pass full-length DesK were performed [15]. Nevertheless, these studies are still hindered by the highly hydrophobic nature of the protein, an issue shared with most transmembrane sensors [1]. Here, we designed an in vivo co-expression system that allows us to investigate in a simplified system the structural organisation of transmembrane regions of multi-spanning full-length DesK. We found that TM1 and TM5, which are covalently linked in MS-DesKC, actually interact in the full-length protein. We also explain, for the first time and at molecular level, what kind of interactions prevail among DesK TM helices to transmit the signal inside the cell.

## 2. Materials and Methods

### 2.1. Plasmid and Strain Constructions

The plasmid that allows the simultaneous expression (co-expression) of two DesK variants from the same promoter was called Coexp TM1/TM5. It was built using plasmid pHPKS (Erythromycin and Lincomycin resistant) [16]. Two DesK variants, TM1-DesKC and TM5-DesKC, were cloned in tandem under the same promoter. TM5-DesKC was cloned using BamHI and XbaI sites, and TM1-DesKC was cloned using XbaI and SacI sites. A six His-tag was added to TM1-DesKC (Figure 1B). The two constructions are separated by an RBS site and spacer. The xylose-inducible promoter, Pxyl, was cloned upstream of the constructions to direct transcription upon xylose addition. TM5-DesKC was cloned using two primers that anneal at the N-terminus of TM5 and C-terminus of DesK. TM1-DesKC was cloned using the overlapping PCR method [9]. The resulting plasmids were used to transform CM21 *Bacillus subtilis* (JH642 *amyE::Pdes-lacZ*(Chloramphenicol)*, desKR::Kanamycin, thrC::Pxyl-desR* (Spectinomycin)) [6]. This strain harbors a deletion in the *desK* gene, and thus will be named as *∆desk* along the text. This strain contains a transcriptional fusion in which the reporter gene *β-galactosidase* is introduced after the promoter of the *desaturase* gene, which allows monitoring DesK kinase activity. QuickChange mutagenesis (Stratagene, La Jolla, CA, USA) was performed to introduce the following mutations in either TM1-DesKC, TM5-DesKC or DesK: TM1-Y17A, TM1-T20A, TM1-Y28A, TM1-DesKC-H188V, TM5-T140A, TM5-S143A, TM5-S150A, TM5-DesKC-H188V, DesK-Y17A, DesK-T20A, DesK-Y28A, DesK-S143A and DesK-S150A (sequences are described in Appendix A. All mutations were confirmed by DNA sequence analysis (MACROGEN, Seoul, Korea), and are listed in Appendix A. Strains are available upon request.

### 2.2. Growth Conditions and Kinase Activity Measurements

For β-galactosidase measurements, the *B. subtilis* JH642- CM21 cells complemented with plasmids encoding DesK variants were grown at 37 °C up to an OD_550_ = 0.3, and then cultures were divided into two: one aliquot remained at 37 °C and the other was transferred to 25 °C. The medium used was made up of Spizizen salts (K_2_HPO_4_ 14.8 g/L; KH_2_PO_4_ 5.4 g/L; (NH_4_)_2_SO_4_ 2 g/L; tri-Sodium citrate·2H_2_O 1.9 g/L; and MgSO_4_·7H_2_O 0.2 g/L) supplemented with 0.1% glycerol, 50 μg/mL each tryptophan, phenylalanine and threonine, 0.05% casa amino acids, trace elements, Lincomycin 25%, Erythromycin 0.1%, Chloramphenicol 5 μg/mL, Kanamycin 5 μg/mL and Spectinomycin 100 μg/mL. All reagents were purchased from Sigma Aldrich (Brussels, Belgium) with molecular biology purity. To induce the expression of DesK variants, 0.8% xylose was added to the growth medium. The cells were collected by centrifugation. Samples were taken at 1-h intervals after resuspension and assayed for β-galactosidase activity as previously described, the activity was expressed in Miller Units [17], in which Optical Density (O.D) at 420 nm is divided by OD_550_ to become independent of bacterial growth. The results shown the average of three independent experiments and correspond to 2 h after the cold shock.

### 2.3. Western Blot Analysis of Coexp TM1/TM5 Mutants

A Western blot was performed to evaluate expression and membrane integration levels of each Coexpression TM1/TM5 variants. *B. subtilis ∆desk* CM21 cells transformed with plasmids expressing each variant were grown at 37 °C in the presence of 0.8% xylose to an OD_550_ of 1. Membrane and cytoplasmic fractions were separated by ultracentrifugation at 45,000× *g* and analysed by Western blot with an Anti-His antibody.

### 2.4. Statistical Analyses

For each strain, we compared the β-galactosidase activity at 25 °C and at 37 °C measured in Miller units (MU) and performed the ANOVA statistical test to determine if the difference between the two measurements was statistically significant. In all cases, a non-parametric test was applied with R commander software. In addition, the corresponding comparisons tests were performed to determine whether the MU (dependent variable) was significantly different between the different categories (25 °C and 37 °C) of the independent variable “temperature” for each strain considering a *p*-value less than 0.05.

## 3. Results

### 3.1. The Co-Expression of Inactive TM1-DesKC and TM5-DesKC Results in an Active Complex

Based on the fact that the fusion of TM1 and TM5 results in a functional TM segment in MS-DesKC, we hypothesised that TM1 and TM5 may interact in full-length DesK. To test this idea, we designed a plasmid that allows the simultaneous expression (co-expression) of two DesK variants. The variants are expressed from the same Pxyl promoter, as if they were two genes in an operon (Figure 1B). Variant TM1-DesKC was constructed by cloning TM1 (represented as a yellow cylinder in Figure 1C) upstream of the cytoplasmic domain of DesK, DesKC (represented as a green circle). Variant TM5-DesKC was constructed by cloning TM5 (represented as a purple cylinder in Figure 1C) upstream of the same cytoplasmic domain, DesKC.

To test activity, MS-DesKC and co-expression variants were expressed from the Pxyl promoter in a ∆desk strain, CM21 (which has the native desK deleted and contains a chromosomal Pdes::lacZ reporter for DesK activity). As it is shown in Figure 1C, when MS-DesKC is expressed, *β*-galactosidase activity was observed at 25 °C, indicative that MS-DesKC is an active DesK and could phosphorylate DesR.*W*hen TM1-DesKC is expressed alone, activity decreases to background levels. Similarly, when only TM5-DesKC is expressed in a ∆desk strain, activity is almost null (Figure 1C). Nevertheless, when these two inactive constructions, TM1-DesKC and TM5-DesKC, are expressed together using the co-expression system, kinase activity at low temperature is partially recovered (43%, Figure 1C). These results show that when expressed alone, either TM1-DesKC or TM5-DesKC, cannot trigger a genetic response. However, when expressed together, TM1-DesKC and TM5-DesKC can recover functionality, which indicates the formation of a heterodimer sensitive to temperature changes. We called this new active system, Coexp TM1/TM5. 

In order to discern whether both catalytic domains fused either to TM1 or TM5 were equivalent in their capacity to process the input signal, we introduced a mutation in the catalytic residue H188.The cytoplasmic domain has two domains: the catalytic and ATP-binding domain (CA) and the Dimerisation and Histidine phosphotransfer domain (DHp. Figure 1D). It has been shown that the replacement H188V abolishes activity in full-length DesK, which is attributed to the fact that the CA domain of one monomer cannot transphosphorylate His188 in the DHp of the other monomer [10]. Since this mutation unables DesK auto-phosphorylation, and consequently DesR cannot become phosphorylated, the reporter activity falls to background levels. The H188V replacement was introduced alternatively in the catalytic domain of either TM1-DesKC or TM5-DesKC in the co-expression system to evaluate functionality (Figure 1D). When H188V is introduced in TM1-DesKC, the system remains active. However, when H188V is introduced in TM5-DesKC, the system is almost inactive (Figure 1D). This result shows that the Coexp TM1/TM5 is not symmetrical, and that in this system, the catalytic domain (CA) associated to TM1-DesKC is capable of catalysing the phosphorylation of H188 in the DHp of TM5-DesKC. Subsequent transfer of the phosphoryl group from TM5-DesKC to the response regulator allows transmitting the signal. On the contrary, the catalytic domain (CA) associated to TM1-DesKC poorly phosphorylates H188 in the DHp of TM1-DesKC. Since it has been demonstrated that the phosphorylation is strongly influenced by the conformation of DHp domain [7], it is possible that the catalytic domain associated to TM1-DesKC is capable of catalysing the phosphorylation step, although the conformation of phosphorylated TM5-DesKC is not conductive for the subsequent phosphoryl transfer to the response regulator. Western blots show similar levels of expression for these mutants (Appendix A).

Together these in vivo results suggest that the information of a temperature change is transmitted directly from TM1 to TM5 (shown with black arrows in the lower panel of Figure 1D). The fact that the heterodimer composed by TM1-DesKC H188V and TM5-DesKC is active suggests that TM1 is involved in sensing membrane lipid fluidity/thickness, and by interacting with TM5, transmits information to the catalytic domain associated to TM5. Therefore, the information flux in this simplified system might be replicating what happens in full length DesK, in which TM5 has been demonstrated to be capable of defining the final conformation of DesKC [18]. These results show that the phosphorylation occurs in trans in the *Coexp TM1*/*TM5*, and how asymmetry of the dimer interface is a key feature for signal transduction. Our findings agree with the phosphotransferase to phosphatase transition model derived from X-ray structures [7] and with previous Molecular Dynamics experiments [8] and experimental data, which suggest that TM1 and TM5 could harbor key sensor elements to sense membrane properties [9,13,18,19].

### 3.2. H-Bond Interactions between TMS Are Required for Signal Transmission

After demonstrating that co-expressing TM1 and TM5 can reconstitute functionality, we wondered how TM1 and TM5 interact to allow signaling in this new system. It has been demonstrated that the formation of interhelical H-bonds between TMS play a critical role in dimer formation and signaling [18,20]. We therefore identified residues with side chains capable of forming H-bonds in either TM1 or TM5, and mutated them for Alanine. It is worth mentioning that we have previously demonstrated that Alanine mutations in very similar TM constructs do not alter the membrane integration of these proteins [18,19]. For the Alanine substitution, we chose firstly the following amino acids: tyrosine 17, threonine 20 and tyrosine 28 of TM1-DesKC construct. All these amino acids contain hydroxyl groups in their side-chain that would allow them to form H-bonds with other H-bond donors or acceptors. These H-bond forming residues locate on the same face of TM1, considering 3–4 residues per helical turn. The individual mutations Y17A, T20A and Y28A were introduced in three different TM1-DesKC constructs of the Coexp TM1/TM5. If these residues were required for interactions with the other TMS, their replacement would cause a decrease in activity. TM1-DesKC variants with the individual replacements Y17A, T20A or Y28A were expressed together with TM5-DesKC in the *∆desk* CM21 strain, using the Coexp TM1/TM5. These three TM1 mutants resulted in a ~77% decrease of the kinase activity regarding Coexp TM1/TM5. We propose that these H-bond forming residues may be interacting with H-bond partner residues in TM5. To test this idea, we mutated the following H-bond forming residues in TM5-DesKC: threonine 140, serine 143 and serine 150, which locate on the same face of TM5. Similarly, the individual mutations T140A, S143A and S150A were introduced in three different TM5-DesKC constructs of the Coexp TM1/TM5. When expressed together with TM1-DesKC in the *∆desk* CM21 strain, using the Coexp TM1/TM5, these three TM5 mutations result in an 83–86% decreaseof the kinase activity regarding the activity level of Coexp TM1/TM5 without mutations (Figure 2A). A Western blot showing similar levels of expression for all the mutants is shown in Appendix A.

### 3.3. Analysis of H-Bonding in Full Length DesK

Finally, we wanted to study whether the residues identified in the Coexp TM1/TM5 as necessary for functionality are also required in full length DesK. For this, we designed a series of DesK mutants with individual replacements in H-bond forming residues in either TM1 or TM5. Each full-length DesK variant was expressed in the *∆desK strain, CM21*). Individual DesK mutants in TM1 (Y17A and Y28A) resulted in a 95% decrease of activity (Figure 3). Mutant T20A resulted in only a 57% decrease in activity, which suggests that this particular position of the helix is less relevant for signal transduction. Similarly, residues S143 and S150, which are located in TM5, were changed to Alanine. These individual replacements also decreased DesK activity by 96% (Figure 3).

## 4. Discussion

Extensive efforts and resources have been destined to study transmembrane kinases; nevertheless, many questions remain about the structural mechanism by which signals pass from the extracellular sensors to the cytoplasmic catalytic domain. There is no crystal structure for any TM in this family, and the NMR structures for the monomeric transmembrane units from ArcB, QseC and KdpD [22] have been solved in detergent micelles at high temperature; however, these structures have limited utility because the dimer interface is not obvious [23].

A pioneering work that shed light on the structural details of TMS interactions showed through an Alanine-scanning study, that Glycine contribute more than others residues to the formation of Glycophorin dimer. This key concept revealed that TMSs have “preferred” interaction interfaces [24]. This was the first discovery of a motif that could confer interaction specificity to a transmembrane helix, and gave rise to the hypothesis that ligand binding to the extracellular domain of a receptor brings monomers together to form an active, signaling-competent dimer. 

Only recently, attention has been paid to specific critical residues in TM helices. For instance, it was reported that Glycine residues play a leading role in TM α-helices, which are far from being rigid; their helical elements are interspersed with flexible joints, which enlarges the possible conformations [1]. They permit enhanced tertiary structural interactions from an increased surface area between helices, thus providing enhanced tertiary structural stability that would otherwise be very limited [25]. One model that illustrates this is the Escherichia coli nitrate/nitrite sensor HK NarQ, in which the conformational changes in the TM domain are facilitated by kinks in the TM helices, where Glycine residues are positioned close. Moreover, glycine residues are also known to be important for TM helix packing by allowing the close approach of the helical backbones [26].

Other motifs proposed to mediate TMS interactions are Aromatic-xx-Aromatic [27], Glutamine-XX-Serine motifs and Leucine zippers [28,29]. These non-covalent interactions follow the same logic: their side-chains promote an interacting interface which tends to form dimers or high-order clusters [30]. Therefore, since TMS interactions are critical to function, understanding the biophysical bases that underlie these interactions is key to comprehend the mechanism of action of these molecular machines. Another interesting example of a polar residue playing a critical role in signal transduction is Asn 202 in the core of the second TM segment of PhoQ. Its replacement to a polar or charged residue abolished the kinase activity, but replacements to other conservative polar residues do not affect signaling. Remarkably, this residue can be moved up or down a turn, or even moved to a neighboring TM helix without significant perturbations to the function [31,32].

The role of hydrogen bonds in the activity of DesK constructs containing one transmembrane segment was initially demonstrated by placing H-bond residues in different faces of the transmembrane helix [18]. The introduction of Asn, a strong hydrogen bond residue, at a particular face of the TM helix induced dimerisation through that particular face, which was detected by cysteine-crosslinking and immune-detection. The activity of the sensor could be manipulated changing the transmembrane dimerisation interface driven by H-bonds.

In this research, we study the *Bacillus subtilis* Histidine Kinase DesK to address how TMSs’ interactions contribute to activity, and found that interactions between TM1 and TM5 are key to signaling, and that these interactions are mainly inter-helical H-bonds. We used a strategy based on vector co-expression of two different TMs fused to the cytoplasmic portion of DesK, which includes the intact catalytic and dimerisation domains (CA and DHp), to test signal transduction via TM-TM interaction. If the interaction is successful, DesK will phosphorylate DesR, which would result in the expression of the promotor *des-lacZ*. This system is not as efficient as the wild type protein, nor the MS-DesKC, and therefore the activities are low. The difference in activities could be the result of multiple factors: the proportion of functional heterodimers (TM1-DesKC/ TM5-DesKC) is low compared to non-functional homodimers (TM1-DesKC/TM1 DesKC and TM5-DesKC/TM5-DesKC). Other factors that may decrease activity in the co-expression system is that the dimer interface in the heterodimer TM1-DesKC/TM5-DesKC is not the most appropriate for sensing or transmission. The proposed TM dimer interface depicted in Figure 2B is defined by Y17-T140, T20-S143 and Y28-S150 in the Coexp TM1/TM5. These interactions are absent in the TM dimer interface in MS-DesKC, which lacks the C-terminus of TM1 and where the dimerisation would be driven by T140-T140, S143-S143 and S150-S150. Therefore, MS-DesKC and Coexp TM1/TM5 do not have the very same orientation nor H-bond contacts, even though the recovery of activity and regulation by the introduction of a TM helix with H-bond residues suggests that H-bond-driven inter-helical interactions play a key role in signal transmission. This allows information to flow from TM1 to TM5 through intermolecular bonds, and contribute to position the cytoplasmic domain in a conformation compatible with catalysis. We propose that the activation mechanism of DesK involves the stabilisation of H-bond interactions between TMS. These transmembrane interhelical H-bonds may be required to position the cytoplasmic catalytic domain in a competent state. According to these and previous findings [8], we propose that at higher temperatures, the membrane is more fluid and hydrated (and that in this environment, the strength of H-bonds is weakened by water screening); while at lower temperatures, the membrane is more rigid and dehydrated, decreasing the dielectric constant and stabilising polar/ electrostatic interactions, which, as shown here, are required for DesK signaling. 

Hydrophilic bonds might be key signal transmitters not only to *B. subtilis* DesK but also to many other Histidine-Kinases. We decided to examine whether the pattern of H-bond forming residues found in TM1 and TM5 of DesK was reproducible in other Histidine Kinases. For that, we first attempted to identify DesK homologs using the TM region as query in a BLAST search. When performing a sequence alignment by BLAST, more than a first thousand nonredundant sequences are homologous to DesK within the genus Bacillus with an identity percentage greater than 90%. Conserved residues of the described pattern are found in all the analysed sequences (for example, *Bacillus halotolerans* DesK, *Bacillus intestinalis* DesK, *Bacillus spizizenii* DesK). To investigate the presence of these residues in Histidine Kinase of putative two-component systems from other genera, *Bacillus* genomes were excluded. In this case, all the sequences found present identity values less than 75%; however, these TM segments display two or three of the residues identified as H-bond-forming residues in the same positions (for example HK from *Mycobacteroides abscessus, Terrabacteria sp, Telluribacter sp and Arthrobacter citreus*, Table 1). Besides, other polytopic sensor proteins as Kcnk1, TRPA1 or Trpv5 also have hydrophilic residues in their TMS that may be interacting with partners in other TMS. For example, Y30-T145 in K2P1 (PDB id 3ukm), N739-N776 in TRPA1 (PDB id 3j9p) and E389-R443 in Trpv5 (PDB id 6dmr) are within H-bond distance, as it can be seen in their crystal structures. Furthermore, Protein–protein interactions within the membrane are involved in many vital cellular processes [30] and mutations in these residues are associated with disease. For example, the introduction of a polar residue (I441T) in the TMS of the Thiamine transporter 1 in *Homo sapiens* leads to Anemia [33] and the mutation G177R/E in the UbiA prenyltransferase domain-containing protein 1, *Homo sapiens*, cause the rare Schnyder crystalline corneal dystrophy [34]. These evidential facts strongly suggest that membrane proteins can be regulated by hydrophilic interactions among their TMS, and changes in these interactions may lead to alterations in the conformation of the catalytic domain. 

To conclude, we anticipate that our findings will inspire further structural experimental and computational studies towards the understanding of the mechanistic details that H-bonds play in signal transduction across the membrane. It will be important to carry out the successful crystallisation for HKs at different temperatures and in their lipid environment rather than in detergents. This is the major challenge to overcome in the future, and it would be indeed a great contribution to demonstrate changes of 3D structure and measure backbone hydrogen bond strengths by hydrogen/deuterium exchange [35].

## Figures and Tables

**Figure 1 biomolecules-11-00938-f001:**
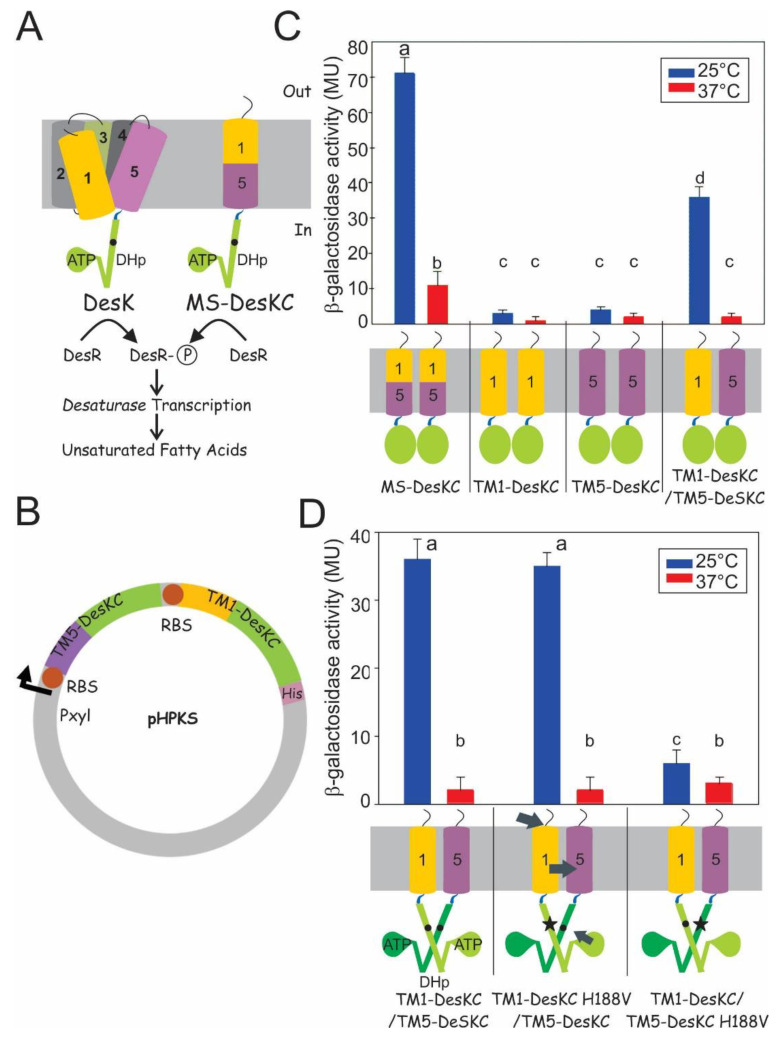
The co-expression of TM1-DesKC and TM5-DesKC restores signal transduction and allows the following flux of information: (**A**) Schema of DesK with five TM helices and the MS-DesKC with one hybrid helix. The proteins are shown as monomers for simplicity. The yellow and violet cylinders represent TM1 and TM5, respectively. The cytoplasmic domain has two domains: the catalytic and ATP binding domain (CA), represented as a green drop with the label ‘ATP’, and the Dimerisation and Histidine phosphotransfer domain (DHp), represented as green cylinders with a black dot highlighting the catalytic H188. When DesK (or MS-DesKC) is active, it phosphorylates the response regulator DesR activating desaturase expression; (**B**) Vector schema of Coexp TM1/TM5 system in pHPKS. The xylose-inducible promoter was cloned upstream of gene constructions coding for TM5-DesKC and TM1-DesKC (for more details see Materials & Methods section); (**C**) Bacillus subtilis ∆desk cells expressing either MS-DesKC, TM1-DesKC, TM5-DesKC or Coexp TM1/TM5 were grown at 37 °C to an OD 525 = 0.3, and then divided in two flasks. One is maintained at 37 °C, and the other transferred to 25 °C. β-galactosidase activity was measured in Miller units (MU). The lower panel shows cartoons of the proteins. The cytoplasmic domain is represented as a green oval for simplicity; and (**D**) Activity and schemas of Coexp TM1/TM5 and its variants with the H188V replacement in either the TM1-DesKC or TM5-DesKC. The black dots represent H188, and the black star highlights the H188V replacement. Each monomer is drawn with different tones. The arrows point out the proposed direction of information flux. Error bars include the standard deviation from at least three independent experiments. There is no significant difference among activities labeled with the same letter. There is significant difference among activities labeled with different letters.

**Figure 2 biomolecules-11-00938-f002:**
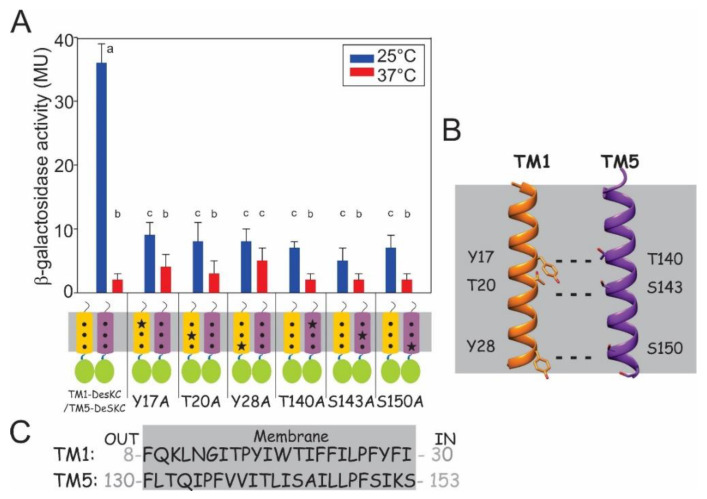
(**A**) TM1-TM5 interhelical hydrogen bonds are required for signaling. *Bacillus ∆desk* cells expressing the co-expression system TM1-DesKC / TM5-DesKC and variants with the indicated mutations were grown as described in Figure 1C. β-galactosidase activity was measured and expressed in Miller units (MU). In the bottom panel, different TMSs are represented as cylinders, the hydrophilic residues (Y17, T20, Y28, T140, S143 and S150) are highlighted with a light-blue dot, and the respective mutations with a black star. Error bars include the standard deviation from at least three independent experiments. There is not significant difference among activities labeled with the same letter. There is significant difference among activities labeled with different letters; (**B**) Molecular graphic of TM1 and TM5 showing the proposed hydrogen bonds between TM1 and TM5. Helices were drawn with UCSF Chimera Software, developed by the Resource for Biocomputing, Visualisation and Informatics at the University of California, San Francisco, with support from NIH P41-GM103311 [21]; and (**C**) Sequence of TM1 and TM5 with the number of the first and last residue of each TMS. The rectangle shows the transmembrane region boundaries according to the bioinformatic software SOSUI and the curated database Uniprot (ID code: O34757).

**Figure 3 biomolecules-11-00938-f003:**
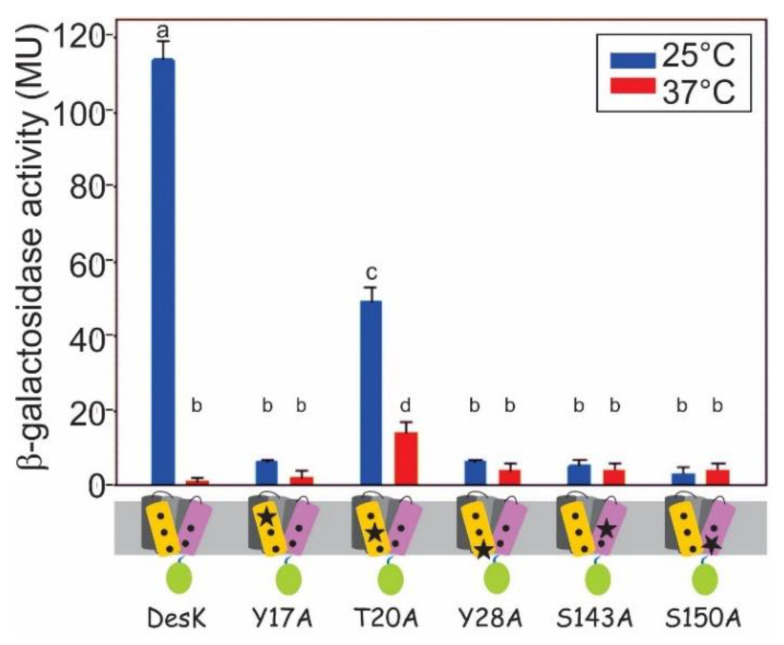
Role of interhelical H-bonds in full length DesK. *B. subtiliss. ∆desk* cells expressing full length DesK and variants with the indicated mutations were grown at 25 °C and 37 °C, as described in Figure 1C. β-galactosidase activity was measured and expressed in Miller Units (MU). Error bars include the standard deviation from at least three independent experiments. There is not significant difference among activities labeled with the same letter. There is significant difference among activities labeled with different letters. In the bottom panel, TMSs are represented as cylinders of different colors and the mutations are highlighted with a black star.

**Table 1 biomolecules-11-00938-t001:** Sequence alignment of TM1 and TM5 using DesK transmembrane domain as query.

NCBI Accession-Microorganism	TM1	TM5	NCBI Accession
DesK *Bacillus subtilis*	LNGITPYIWTIFFILPFYFIW	FFLTQIPFVVITLISAILLPFS	WP_041339839.1
DesK *Bacillus halotolerans*	LNGITPYIWTIFFILPFYFIF	FFLTQIPFVVITLISAILLPFS	WP_105955034.1
DesK *Bacillus intestinalis*	LSGITPYIWTIFFILPFYFIW	FFLTQIPFVVITLISAILLPFS	WP_087986990.1
DesK *Bacillus spizizenii*	LSGITPYIWTIFFILPFYFIW	LFLTQIPFVVITLISAILLPFS	WP_187952809.1
HK *Mycobacteroides abscessus*	LNGISPYIWTTFFILPFYFIF	WFLTQIPFIVITLISAILLPLT	SLA98150.1
HK *Terrabacteria* sp.	NSGISPYIWTILCILPFYFIF	LFLKQLPFVIIIWISVILLPFN	WP_095249154.1
HK *Telluribacter* sp.	STGISPYIWTILGILPFYFIW	FFIKQLPIIVIVWISVILLPFS	WP_207499360.1
HK *Arthrobacter citreus*	STGISPYIWTVLGISPFYFIF	LLIRQLPFIIVTWISVILLPFS	WP_172444440.1
HK RLJ69699.1 *Actinophytocola xinjiangensis*	NHGLSPYVWIFLSILPFYFIF	-FITQLPFVFLSLIAVILLPVY	RLJ69699.1

H-bond residues are highlighted in yellow when exactly the same residue is maintained in that position, or in blue when there is another hydrophilic H-bond residue at that position.

## Data Availability

Not applicable.

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
