# Peer review of "Interhelical H-Bonds Modulate the Activity of a Polytopic Transmembrane Kinase"

_biomolecules, 2021, doi:10.3390/biom11070938_

Round 1

Reviewer 1 Report

Overall, the manuscript presented by Juan Cruz Almada et al., is a nice piece of work. It uses a strategy of vector co-expression of two different TMs fused to the cytoplasmic portion of DesK to test signal transduction via TM-TM interaction. If interaction is successful, the phosphorelay DesK-DesR takes place and the amount of DesR-phosphorylation is measured as a reporter activity. Also, the authors use a battery of mutants to demonstrate that TMs communicate through interactions that involve polar residues. My only concern is that, in general, the Miller units obtained in the assay are low. As different yields in protein expression can have an impact in the activity observed, it would be desirable to compare protein expression for all variants, maybe through a western blot experiment. In any case, I believe that the manuscript is suitable for publication after the following minor changes are taken into account.

  1. Line 25. Write histidine kinase separated, and not as histidine-kinase.

  1. Line 42. Indicate that DesK is a histidine kinase, as you did in the abstract.

  1. Line 45. The paragraph that finishes at this line indicates looking at Fig 1A. However, what is mentioned in the paragraph does not correlate with the image. Thus, Fig. 1A should be moved to an appropriate part such as the end of paragraph at line 47 and at line 50.

  1. Line 59. The authors assign Fig. 1B to the construction but in Fig. 1, panel B is the activity experiment. Please change Fig. 1C to Fig 1B.

  1. Line 77 in Materials and methods. Indicate that the Bacillus subtilis strain harbouring a deletion in the desk gene will be named as desk- along the text. Also, it should be written with clarity how the fusion of the reporter gene (beta-galactosidase) and promoter of desaturase have been designed, that is, if the reporter gene was introduced after the promoter of the desaturase gene?

I would recommend using the ∆desk (genes are in italics) acronym instead of desk- to indicate the strain that has desk gene deleted from the genome of B. subtilis.

The first time that the author use Bacillus subtilis strain should provide the complete name in italics. But the subsequent times, the author should use B. subtilis in italics.

Line 78. beta-galactosidase and desaturase genes in italics.  

  1. Line 89-90. Could you explain in the text why it is necessary to add 5 antibiotics?

  1. Line 104. Results section 3.1

The authors start with a sentence stating that the fusion TM1-TM5 reproduces a functional DesK before explaining the experiment. First, and as it is explained in Line 53-57, the authors should demonstrate that expression of MS-DesKC produces an active protein. Thus, I recommend including previously the following sentence “Experiments were performed to test expression of MS-DesKC variant in the desk- strain. As it is shown in Fig. 1C (it would be 1B in your manuscript before changing panels) beta-galactosidase activity was observed at 25°C, indicative that MS-DesKC was an active DesK and could phosphorylate DesR”.

As DesKC can dimerize in the absence of TM domains, it would be interesting to add the expression of DesKC alone in desk- cells at 25 and 37 °C as a control of no activity, just to make sure that there is no signal transduction in the absence of TMs. Do the authors have any information about that? If yes, please include it in the manuscript.

I believe that the Miller units obtained in the assay are low in general and different yields in protein expression can have an impact in the activity observed. As the plasmid for TM1-DesKC has Histag, could you do a western blot with anti-His Ab to compare if the expression of the proteins expressed are similar in all cases? If you already have information about that, it would be very useful to include it or mention it.

  1. Line 117-118. Indicate that the co-expression of the inactive variants in desk- strain recovers 50% of the activity observed for the MS-DesKC variant.

  1. Line 129-130. Indicate that the mutation H188V unables DesK autophosphorylation and stabilizes the phosphatase conformation of DesK, thus, DesR cannot become phosphorylated and so reporter activity cannot be observed.

Please change DHP to DHp.

  1. Figure caption 1.

-  Change section B) for C) as mentioned in 4.

- Line 155. Bacillus DesK- cells it is not properly written. See 5.

- Line 158-161. Results should not be written in the figure caption and should be moved to results section.  

  1. Line 187. Please indicate for clarity that the individual mutations Y17A, T20A and Y28A where located in the TM1-DesKC construct.

  1. Section 3.2. It would be desirable to give a quantitative value for the activity of the mutants co-expressed versus the WT. As I can infer from the graphic in Fig. 2, about a 20-30% of the activity is obtained for the mutants.

  1. Results section 3.3

- Line 216. The authors indicate that the mutant T20A results in a significant decrease in activity, but this mutant shows 50% of the activity for the WT. The authors should mention it in the manuscript. In general, the authors should make an effort to quantitatively express differences in the activities along the manuscript. 

-The authors do not test the mutant T140A in the DesK full length protein, could you explain why?

  1. Line 307. The citation does not contain the journal, please add it.

Author Response

Overall, the manuscript presented by Juan Cruz Almada et al., is a nice piece of work. It uses a strategy of vector co-expression of two different TMs fused to the cytoplasmic portion of DesK to test signal transduction via TM-TM interaction. If interaction is successful, the phosphorelay DesK-DesR takes place and the amount of DesR-phosphorylation is measured as a reporter activity. Also, the authors use a battery of mutants to demonstrate that TMs communicate through interactions that involve polar residues. My only concern is that, in general, the Miller units obtained in the assay are low. As different yields in protein expression can have an impact in the activity observed, it would be desirable to compare protein expression for all variants, maybe through a western blot experiment. In any case, I believe that the manuscript is suitable for publication after the following minor changes are taken into account.

Thank you for your point of view. We agree with the reviewer, the activities are very low, and to co-relate with expression levels, we performed a Western blot, which shows similar levels for all co-expression variants (SUpp. Fig 1).

The lines corresponding to modifications introduced in the new version refer to the text with active track changes, as suggested by the editor.

  1. Line 25. Write histidine kinase separated, and not as histidine-kinase.

The correction was included (line 21, 32)

  1. Line 42. Indicate that DesK is a histidine kinase, as you did in the abstract.

The introduction was modified, and the correction included, line 48.

  1. Line 45. The paragraph that finishes at this line indicates looking at Fig 1A. However, what is mentioned in the paragraph does not correlate with the image. Thus, Fig. 1A should be moved to an appropriate part such as the end of paragraph at line 47 and at line 50.

The text was modified accordingly, line 50.

  1. Line 59. The authors assign Fig. 1B to the construction but in Fig. 1, panel B is the activity experiment. Please change Fig. 1C to Fig 1B.

We changed  the figure and legend accordingly.

  1. Line 77 in Materials and methods. Indicate that the Bacillus subtilis strain harbouring a deletion in the desk gene will be named as desk- along the text. Also, it should be written with clarity how the fusion of the reporter gene (beta-galactosidase) and promoter of desaturase have been designed, that is, if the reporter gene was introduced after the promoter of the desaturase gene?

We introduced the modifications about the strain along the texto. An explanation for the construction of the promoter-reporter gene is described in lines 117-121.

I would recommend using the ∆desk (genes are in italics) acronym instead of desk- to indicate the strain that has desk gene deleted from the genome of B. subtilis.

The modifications were included.

The first time that the author use Bacillus subtilis strain should provide the complete name in italics. But the subsequent times, the author should use B. subtilis in italics.

 The modifications were included

Line 78. beta-galactosidase and desaturase genes in italics.  

The modifications were included

  1. Line 89-90. Could you explain in the text why it is necessary to add 5 antibiotics?

 We describe the strain used, CM21 with the complete genotype. The antibiotic resistance gene, kanamycin, was used to interrupt DesK. Since Desk and the response regulator DesR constitute an operon, when DesK is mutated, the expression of the response regulator, which is coded downstream of DesK, is hampered due to a polar effect. To test the effect of DesK mutants, DesR must be present in the cell. So, the gene coding for the response regulator is expressed under the xylose promoter in the threonine locus (construction carries spectinomycin gene for selection), the reporter Pdes-lacZ carries chloramphenicol for selection, and the plasmid carrying the DesK variant has the gene coding for Macrolides (Erythromycin-Lyncomycin). In the Materials and Method section we include the complete genotype of CM21, a brief description and a reference in which the construction of the strain is detailed, lines 108, and lines 116-118.

  1. Line 104. Results section 3.1

The authors start with a sentence stating that the fusion TM1-TM5 reproduces a functional DesK before explaining the experiment.

First, and as it is explained in Line 53-57, the authors should demonstrate that expression of MS-DesKC produces an active protein. Thus, I recommend including previously the following sentence “Experiments were performed to test expression of MS-DesKC variant in the desk- strain. As it is shown in Fig. 1C (it would be 1B in your manuscript before changing panels) beta-galactosidase activity was observed at 25°C, indicative that MS-DesKC was an active DesK and could phosphorylate DesR”.

We introduced the suggestion, lines 170-174. 

As DesKC can dimerize in the absence of TM domains, it would be interesting to add the expression of DesKC alone in desk- cells at 25 and 37 °C as a control of no activity, just to make sure that there is no signal transduction in the absence of TMs. Do the authors have any information about that? If yes, please include it in the manuscript.

It is true that DesKC dimerizes in the absence of TM domains, nevertheless, the activity of DesKC is constitutive and very high (1000 units), so including this DesK variant in the graph will make the rest too difficult to see. We included a comment and a reference where the activity of DesKC is measured, lines 53-55.

I believe that the Miller units obtained in the assay are low in general and different yields in protein expression can have an impact in the activity observed. As the plasmid for TM1-DesKC has Histag, could you do a western blot with anti-His Ab to compare if the expression of the proteins expressed are similar in all cases? If you already have information about that, it would be very useful to include it or mention it.

Yes, in the new version of the paper a anti-his immunoblot is included. It shows the levels are low, although similar for all co-expression variants (Supp. Fig 1).

  1. Line 117-118. Indicate that the co-expression of the inactive variants in desk- strain recovers 50% of the activity observed for the MS-DesKC variant.

The suggestion was included, line 179.

  1. Line 129-130. Indicate that the mutation H188V unables DesK autophosphorylation and stabilizes the phosphatase conformation of DesK, thus, DesR cannot become phosphorylated and so reporter activity cannot be observed.

The suggestion was included (lines 190)

Please change DHP to DHp.

It was corrected alogn the text.

  1. Figure caption 1.

-  Change section B) for C) as mentioned in 4.

The change was included

- Line 155. Bacillus DesK- cells it is not properly written. See 5.

It was corrected

- Line 158-161. Results should not be written in the figure caption and should be moved to results section.  

The correction was included, lines 224-253.

  1. Line 187. Please indicate for clarity that the individual mutations Y17A, T20A and Y28A where located in the TM1-DesKC construct.

We included the suggestion, lines 263, and 266-268.

  1. Section 3.2. It would be desirable to give a quantitative value for the activity of the mutants co-expressed versus the WT. As I can infer from the graphic in Fig. 2, about a 20-30% of the activity is obtained for the mutants.

Quantifications were performed for all mutants.

  1. Results section 3.3

- Line 216. The authors indicate that the mutant T20A results in a significant decrease in activity, but this mutant shows 50% of the activity for the WT. The authors should mention it in the manuscript. In general, the authors should make an effort to quantitatively express differences in the activities along the manuscript. 

We mention the behavior of mutant T20A in the manuscript (line 307), and quantitatively express differences in activities along the MS.

-The authors do not test the mutant T140A in the DesK full length protein, could you explain why?

At the beginning, our idea was to test only few mutants in full length DesK, to see if the general behavior was maintained in the two systems. Then we introduced more and more mutants, but not all of them. We agree having the same mutations in both systems would be more appropriate.

  1. Line 307. The citation does not contain the journal, please add it.

It was included.

Reviewer 2 Report

This manuscript needs major revisions. It is missing important experimental results and details of methodology that are central to the paper, and both introduction and discussion need detail and extension.

Introduction.

  1. This intro is not sufficient to explain background and significance of this work. This is done only in the conclusive paragraph (line 276).

DesK is described in a context that is detailed and specific, however it does not highlight the significance of this protein study in a broader context, except for saying that signal transduction is a mystery.

Why is studying DesK significant?

How many bacterial proteins share high homology?

Are there putative human analogs?

Line 39 should contain at least one reference on the current knowledge or a recent review on signal transduction. The statement that  it is incomprehensible for most systems is not sufficient (and quite dismissive to everyone who is making progress on understanding signal transduction)

  1. Overall the Intro is unclear about what the article is about, it is as informative as the abstract. The only information is Figure 1B. The introduction text should introduce the proteins that are expressed even if in broad lines.

Line 57. What does the rest of the protein do? There is no description of the putative  function of the rest of the protein, and it is not clear why it would be produced as a 5TM spanning protein or as a dimer. The authors cannot assume every reader knows DesK literature in detail.

Figures. All figures appear blurry on the reviewer preprint. Perhaps figure format is incorrect. Figure 1D appears truncated on the bottom.

Figure 1B legend line 158-160: Figure 1B is interpreted by saying that there is no significant difference between MSDesK at 37 degrees and the other measurements at 37 degrees. However, even with the error bars, the MSDesK activity at 37 is significantly higher than the others, and looks over twice that for Coexp at 37C, so this statement does not seem reflected in the data. Activities of Coexp at each temperature are about half that for MS DesK. This huge disparity should be discussed. Lines 53-66. Experimental details and/or references for galactosidase activity are missing.

Figure 1D schematic is very unclear and it appears truncated at the bottom, so the ATP biding domain seems like a separate molecule. Everywhere else the cytoplasmic domain is represented as a green oval. Is it the same unit as this stick representation? The two green tones are hardly visible.

Legend lines 170-175 are unclear. Between what constructs are differences found to be not significant? Coexp TM1/TM5 at 25C is significantly different from the other constructs at the same temperature. This needs to be re-written more clearly and explicitly, specifying what is compared to what.

Dimerization: a brief mention of dimerization is provided in line 53, but it is not clear what was previous experimental proof that DesK is a dimer. This should be stated clearly in the into or in the discussion. What is the oligomeric state of all the other constructs described here? What is the proof that MC-DesKC TM1 or TM5 form dimers? Can the overall lower activity of Coexp TM1/TM5 (Fig1B) compared to the MSDesKC be explained with a mixture of TM1/TM1, TM1/TM5,TM5/TM5 monomers/dimers present in equilibrium at different ratios?

Materials and Methods:

79-81: Strains, full sequences and detailed construction methods are the core of this work, and it is not sufficient to say that they are available upon request. This manuscript needs a detailed description of these details, which can be described here or in a supplementary section. This cannot be 'optional' upon request.

This  methods section is missing a description of  how expression levels were verified/measured. This is necessary since the whole work hinges on the expression in vivo. Again, this can be described here on in a supplementary section, but it is necessary.

-Throughout the manuscript the word is referred to with slightly different notations with different letters capitalized, e.g, line 77 desk-; line 91 Desk; line 115 desK-

-Lines 82-92. This paragraph contains several typographic and editing errors. A few are:

86: extra space

84+ 87 + 88: incorrect use of comma, in 0,3, 0,1% and 0,05%. Should be a period.

88 ug, fix with Greek letter notation for micro

Results:

The residue 188 mutated in TM1 is not naturally on the first TM, which makes this transduction model somewhat artificial and not particularly convincing. The authors should better describe the significance of the TM1 construct and how it translates in the transduction directionality model.

In Results, line 114 ‘we tested expression”, however, tests are not included here or in the methods section. They should be included since they are central to this work.

Lines 181, please substitute construct instead of constructions, and correct typo no->not

Line 200-beta letter missing

Reference for Chimera software missing

The paper’s conclusion is a detailed structural model based on mutations and functional measurements. Actual structural/physico-chemical measurements to prove the presence of the postulated hydrogen bonds are still missing. In Discussion, the authors should discuss how this model could potentially be conclusively demonstrated or addressed in the future.

Author Response

Comments and Suggestions for Authors

This manuscript needs major revisions. It is missing important experimental results and details of methodology that are central to the paper, and both introduction and discussion need detail and extension.

Thank you for the comment. We agree in that important issues were missing along the paper. The lines corresponding to modifications introduced in the new version refer to the text with active track changes, as suggested by the editor.

Introduction.

This intro is not sufficient to explain background and significance of this work. This is done only in the conclusive paragraph (line 276).

The introduction was re-written entirely to explain Histidine-kinases in general as well as the DesK background (lines 35-46, 88-99).

DesK is described in a context that is detailed and specific, however it does not highlight the significance of this protein study in a broader context, except for saying that signal transduction is a mystery. Why is studying DesK significant?

We agree with the reviewer. The introduction was modified to include background and significance of DesK in a broader context.

In the new version we explain why studying DesK is significant in the field of signal transduction (Lines 47-55)

How many bacterial proteins share high homology?

When performing a sequence alignment using BLAST and the transmembrane domain as the query, more than a first thousand sequences correspond to Desk variants of the genus Bacillus. Sequences from other genera (Mycobacteroides abscessus,Terrabacteria sp, Telluribacter sp, Arthrobacter citreus) only emerged when Bacillus is excluded. A paragraph including this analysis is now included in lines 415-426.

Are there putative human analogues?

We looked for putative analogs, but unrelated proteins, like FLD-1 and TLCD1/2 (not homologous to DesK) have been proposed to be involved in fluidity sensing in humans.

Line 39 should contain at least one reference on the current knowledge or a recent review on signal transduction. The statement that  it is incomprehensible for most systems is not sufficient (and quite dismissive to everyone who is making progress on understanding signal transduction).

We agree with the reviewer. We briefly describe the current knowledge for TCS  signal transduction and include references: lines 39-40 (introduction) and line 329 (discussion).

Overall the Intro is unclear about what the article is about, it is as informative as the abstract. The only information is Figure 1B. The introduction text should introduce the proteins that are expressed even if in broad lines

We improved the whole introduction by including additional information about how previous findings motivated the research of this work. We also included more details and a more complete introduction of the proteins used in this study (lines 35-104).

Line 57. What does the rest of the protein do? There is no description of the putative  function of the rest of the protein, and it is not clear why it would be produced as a 5TM spanning protein or as a dimer.

We don´t know what the rest of the protein does.

The regulation of the activity is better in wild type DesK than in MS-DesK: DesK is more active in the presence of the signal (25°C) and background levels are lower in the absence of the signal (37°C). This difference in regulation suggests that the other TMS do play a role in fine tunning sensing. Nonetheless, it is also possible that the other TM helices play a role detecting unknown signals. We are working trying to understand the role of the other TMS. A sentence including these ideas is included in lines 85-88.

In the new version, a new paragraph describing that almost all of the HK that are characterized structurally are homodimeric (line 39) and explain what is known for DesK dimerization (lines 72-74, and 368-370)

The authors cannot assume every reader knows DesK literature in detail.

We agree, and made an effort to synthetize the main findings in DesK structure and sensing/ signaling (new introduction, lines 47-99).

Figures. All figures appear blurry on the reviewer preprint. Perhaps figure format is incorrect. Figure 1D appears truncated on the bottom.

We checked the format of the figures (they are new) and corrected Fig.1D. We hope this time they do not look blurry.

Figure 1B legend line 158-160: Figure 1B is interpreted by saying that there is no significant difference between MSDesK at 37 degrees and the other measurements at 37 degrees. However, even with the error bars, the MSDesK activity at 37 is significantly higher than the others, and looks over twice that for Coexp at 37C, so this statement does not seem reflected in the data.

This was a mistake. Sorry for this. There is a difference between MS-DesK at 37 degrees and the other measurements at 37 degrees and described in line 86 and lines 248-250.

 Activities of Coexp at each temperature are about half that for MS DesK. This huge disparity should be discussed:

We notice that activities of the Co-exp TM1/TM5 at each temperature are about half that for MS DesK. The difference in activities could be the result of multiple factors: the proportion of functional heterodimers (TM1-DesKC/ TM5-DesKC) is low compared to non-functional homodimers (TM1-DesKC/TM1 DesKC and TM5-DesKC/TM5-DesKC), as this reviewer noticed below. Other factor that may decrease activity in the co-expression system is that the dimer interface in the heterodimer TM1-DesKC/TM5-DesKC is not be the most appropriate to sense and transmit the signal. We mention this in lines 390-400.

 Lines 53-66. Experimental details and/or references for galactosidase activity are missing.

We included experimental details and reference for this assay (lines 138-143).

Figure 1D schematic is very unclear and it appears truncated at the bottom, so the ATP biding domain seems like a separate molecule. Everywhere else the cytoplasmic domain is represented as a green oval. Is it the same unit as this stick representation? The two green tones are hardly visible.

We improved the figure 1A and 1D: we avoided the truncated appearance, and changed colors and shapes. The green oval is the same unit as the stick representation, which is now also used and explained in Fig. 1A-D.

Legend lines 170-175 are unclear. Between what constructs are differences found to be not significant? Coexp TM1/TM5 at 25C is significantly different from the other constructs at the same temperature.

This needs to be re-written more clearly and explicitly, specifying what is compared to what.

The statistical analysis was re-written to explain better the differences in activities among the constructs (lines 248-250, Fig.1). Also for the other figures 291-293 (Fig 2), and 315-317 (Fig.3)

Dimerization: a brief mention of dimerization is provided in line 53, but it is not clear what was previous experimental proof that DesK is a dimer. This should be stated clearly in the intro or in the discussion. What is the oligomeric state of all the other constructs described here?

 What is the proof that MC-DesKC TM1 or TM5 form dimers?

In the new version, we describe that almost all histidine-kinases are homodimers (except two), lines 39-40, and mention the experiments/results that show that DesK is also a dimer (lines 368-370). The known dimerization domain is the cytoplasmic dimerization and phsophotransfer domain ( DHp), as revealed in several x-ray structures (now described in line 71). This  domain is present (and not modified) in all the contructions; so, we assume that all DesK variants will dimerize through the DHp domain (now mentioned in lines 386-387).

Since a DesK monomer transphosphorylate the other monomer within the dimer, we deduce activity is only recovered when a functional dimer (or heterodimer) has been formed.

 Can the overall lower activity of Coexp TM1/TM5 (Fig1B) compared to the MSDesKC be explained with a mixture of TM1/TM1, TM1/TM5,TM5/TM5 monomers/dimers present in equilibrium at different ratios?

Exactly. This perceptive comment is now included in the discussion , lines 390-400.

Materials and Methods:

79-81: Strains, full sequences and detailed construction methods are the core of this work, and it is not sufficient to say that they are available upon request. This manuscript needs a detailed description of these details, which can be described here or in a supplementary section. This cannot be 'optional' upon request.

A detailed description of strains, sequences and methods are now described in a supplementary section (strain genotype, method details) line 106-127

This  methods section is missing a description of  how expression levels were verified/measured. This is necessary since the whole work hinges on the expression in vivo. Again, this can be described here on in a supplementary section, but it is necessary.

In the new version, we included a Western blot showing similar levels for all co-expression variants (SUpp. Fig 1).

-Throughout the manuscript the word is referred to with slightly different notations with different letters capitalized, e.g, line 77 desk-; line 91 Desk; line 115 desK-

Corrections were introduced along the MS.

-Lines 82-92. This paragraph contains several typographic and editing errors. A few are:

86: extra space

84+ 87 + 88: incorrect use of comma, in 0,3, 0,1% and 0,05%. Should be a period.

88 ug, fix with Greek letter notation for micro

The corrections were introduced (lines 128-143)

Results:

The residue 188 mutated in TM1 is not naturally on the first TM, which makes this transduction model somewhat artificial and not particularly convincing. The authors should better describe the significance of the TM1 construct and how it translates in the transduction directionality model.

It is true that the co-expression system is artificial. We describe the significance of the TM1 construct in lines 215-217 and 399-404.

In Results, line 114 ‘we tested expression”, however, tests are not included here or in the methods section. They should be included since they are central to this work.

We re-wrote the sentence (lines 170). Expression levels were checked by Western blots (Fig Supp1)

Lines 181, please substitute construct instead of constructions, and correct typo no->not

Changes were introduced, line 261.

Line 200-beta letter missing

The correction was introduced.

Reference for Chimera software missing

The reference is included, line 294-297.

The paper’s conclusion is a detailed structural model based on mutations and functional measurements. Actual structural/physico-chemical measurements to prove the presence of the postulated hydrogen bonds are still missing. In Discussion, the authors should discuss how this model could potentially be conclusively demonstrated or addressed in the future.

It is indeed the next step. It will require the availability of a 3D structure with a resolution sufficient to estimate the distances between side chains of the residues located in the transmembrane domains and consequently the existence of hydrogen bonds. It will be important to carry out the successful crystallisation at different temperatures and in a lipid environment rather in a detergent. Cryo-EM would be a way to address the question elegantly. Equilibrium hydrogen/deuterium could be employed to measure backbone hydrogen bond strengths ( Z.Cao et al JACS 139;10742- 2017).These are the major challenges to overcome in the future and it would be indeed a  great contribution to demonstrate  changes of  3D structure resulting from a change of the protein lipid environment but it is  beyond the scope of the present work. This is briefly mentioned in the very last paragraph of the text (line 439)

Reviewer 3 Report

In their manuscript, Almada and colleagues present a series of experiments with a bacterial transmembrane histidine kinase, DesK, which is a sensor for membrane fluidity. Histidine kinases are important bacterial signaling proteins, which are at the same time very hard to study, and many questions still remain unanswered. Consequenly, the study provides a welcome contribution and important insights.

In particular, the authors generate first a simplified system, where the cytoplasmic domains of DesK, having 5 transmembrane (TM) helices, are fused either to helix 1 or to helix 5. The authors show that the homodimers of TM1 or TM5 constructs have strongly diminished activity, whereas co-expressed heterodimers have an activity close to that of the original construct. The authors then notice that the hydrogen bond-forming amino acids are important for activity of the protein, and demonstrate this also for the full-length protein.

Overall, I think that this is a well done study providing important insights into the function of DesK. The manuscript is clearly written and easy to understand. However, before it can be published, I think that several parts of the discussion can be amended to understand the generality of the findings as described below :

1) It might be interesting to check whether these polar residues are conserved among DesK proteins from different organisms. A simple BLAST search and alignment of 3-5 sequences from representative organisms can be very informative.

2) It would be nice to put the work in context of the data available for other histidine kinases (as reviewed, for example, by Bhate et al., doi:10.1016/j.str.2015.04.002, and Gushchin and Gordeliy, doi: 10.1002/bies.201700197).

For example, in the lines 238-244 the authors discuss the role of glycines in TM helices in general. It is nice, however, it would also be interesting if the authors discussed the role of glycines in histidine kinases, which is much more relevant to the study (see again the Gushchin and Gordeliy review, doi:10.1002/bies.201700197). The same applies to the discussion presented in lines 276-289.

Also, the article 10.1073/pnas.1003166107 explicitly states "Transmembrane polar interactions are required for signaling in the Escherichia coli sensor kinase PhoQ", which is directly relevant for the present manuscript. Articles 10.1073/pnas.1001656107, 10.1016/j.str.2014.04.019, 10.1126/science.aah6345 provide the models of TM domains of histidine kinases, which have many polar interactions.

3) lines 136-137, and also 138-139: I do not agree with these statements and I think that the discussion presented here and also in the lines 140-148 has to be amended. Phosphorylation is strongly influenced by the conformation of DHp domain. It is possible that the catalytic domain associated to TM1-DesKC is capable of catalyzing the phosphorylation step, however the conformation of TM5-DesKC DHp domain is not conductive for phosphorylation. The article by Trajtenberg and colleagues (doi:10.7554/eLife.21422) appears to be particularly relevant for the discussion, especially because it deals with the same kinase that the authors are studying.

4) The original MS-DesKC construct apparently didn't have the interactions depicted in Fig. 2B. However, it is capable of sensing the membrane fluidity. Can the authors provide some comment on this?

Minor issues:

Bacteria and gene names should probably be italicized

lines 35-40: this is generally correct; however, the signal sensed by DesK is not environmental and the sensor domain is not extracellular, so it would be nice to improve this paragraph to cover also DesK, on which the manuscript is focused.

line 45 and elsewhere: DesK has two cytoplasmic domains: DHp and the catalytic domain (usually abbreviated in the histidine kinase literature as CA). This is mentioned correctly in lines 126-128, but not in line 45 or in the abstract, which may be confusing for the reader.

line 48: I suggest replacing "phosphorylation" with "autophosphorylation" here and elsewhere where appropriate. While the phosphorylation occurs in trans, I think it can be said that the whole dimer autophosphorylates

line 68: what is "Coexp TM1/TM5" ? this is the first time it appears in the text

line 115: judging from fig. 1B, the activity is not null, although significantly diminished compared to MS-DesKC. The same is true for lines 189-190: the activity is not lost completely, it is just diminished. Importantly, it is not clear whether this comes from lower expression level or lower activity of kinase per se.

lines 231-237: it would be nice to support each of these statements with appropriate references.

Author Response

In their manuscript, Almada and colleagues present a series of experiments with a bacterial transmembrane histidine kinase, DesK, which is a sensor for membrane fluidity. Histidine kinases are important bacterial signaling proteins, which are at the same time very hard to study, and many questions still remain unanswered. Consequenly, the study provides a welcome contribution and important insights.

In particular, the authors generate first a simplified system, where the cytoplasmic domains of DesK, having 5 transmembrane (TM) helices, are fused either to helix 1 or to helix 5. The authors show that the homodimers of TM1 or TM5 constructs have strongly diminished activity, whereas co-expressed heterodimers have an activity close to that of the original construct. The authors then notice that the hydrogen bond-forming amino acids are important for activity of the protein, and demonstrate this also for the full-length protein.

Overall, I think that this is a well done study providing important insights into the function of DesK. The manuscript is clearly written and easy to understand. However, before it can be published, I think that several parts of the discussion can be amended to understand the generality of the findings as described below :

Thank you for the comments. We agree in that important issues were missing in the paper. The lines corresponding to modifications introduced in the new version refer to the text with active track changes, as suggested by the editor.

  • It might be interesting to check whether these polar residues are conserved among DesK proteins from different organisms. A simple BLAST search and alignment of 3-5 sequences from representative organisms can be very informative.

Thanks for the contribution. It was rewarding to find that the H-bonds we analyze here are conserved in different DesK as well as in other HK. The Discussion now includes a comment and a table showing the conservation of H-bond residues, lines 413-427.

  • It would be nice to put the work in context of the data available for other histidine kinases (as reviewed, for example, by Bhate et al., doi:10.1016/j.str.2015.04.002, and Gushchin and Gordeliy, doi: 10.1002/bies.201700197).

We put the work in a wider context, and these references were very helpful (lines 39-46 and 326-329)

For example, in the lines 238-244 the authors discuss the role of glycines in TM helices in general. It is nice, however, it would also be interesting if the authors discussed the role of glycines in histidine kinases, which is much more relevant to the study (see again the Gushchin and Gordeliy review, doi:10.1002/bies.201700197). The same applies to the discussion presented in lines 276-289.

Also, the article 10.1073/pnas.1003166107 explicitly states "Transmembrane polar interactions are required for signaling in the Escherichia coli sensor kinase PhoQ", which is directly relevant for the present manuscript. Articles 10.1073/pnas.1001656107, 10.1016/j.str.2014.04.019, 10.1126/science.aah6345 provide the models of TM domains of histidine kinases, which have many polar interactions.

In the new version, a description about glycines and H-bond forming resiudes in HK was included in lines 345-354 and 413-427, and the interesting work by Goldberg included in the discussion, lines 360-365.

  • lines 136-137, and also 138-139: I do not agree with these statements and I think that the discussion presented here and also in the lines 140-148 has to be amended. Phosphorylation is strongly influenced by the conformation of DHp domain. It is possible that the catalytic domain associated to TM1-DesKC is capable of catalyzing the phosphorylation step, however the conformation of TM5-DesKC DHp domain is not conductive for phosphorylation. The article by Trajtenberg and colleagues (doi:10.7554/eLife.21422) appears to be particularly relevant for the discussion, especially because it deals with the same kinase that the authors are studying.

This was a mistake. Sorry for this. We amended the new version with this observation. We conclude that the catalytic domain associated to TM1-DesKC is capable of catalyzing the phosphorylation step and include the reference to help the analysis (Trajtenberg), lines 196-206.

4) The original MS-DesKC construct apparently didn't have the interactions depicted in Fig. 2B. However, it is capable of sensing the membrane fluidity. Can the authors provide some comment on this?

We like this inspiring comment. We agree with the reviewer and the new version includes a comment on this difference (lines 395-400).

Minor issues:

Bacteria and gene names should probably be italicized

The corrections were included

lines 35-40: this is generally correct; however, the signal sensed by DesK is not environmental and the sensor domain is not extracellular, so it would be nice to improve this paragraph to cover also DesK, on which the manuscript is focused.

The introduction was improved and now we cover DesK in the general description of HK, lines 35-37, and 47-51.

line 45 and elsewhere: DesK has two cytoplasmic domains: DHp and the catalytic domain (usually abbreviated in the histidine kinase literature as CA). This is mentioned correctly in lines 126-128, but not in line 45 or in the abstract, which may be confusing for the reader.

The correction was included in the new version (lines 42-46). In the Abstract we prefer to talk about the cytoplasmic domain as a localization unit, instead of already talk about the two structural sub-domains.

line 48: I suggest replacing "phosphorylation" with "autophosphorylation" here and elsewhere where appropriate. While the phosphorylation occurs in trans, I think it can be said that the whole dimer autophosphorylates

The suggested changes were introduced

line 68: what is "Coexp TM1/TM5" ? this is the first time it appears in the text

We explain what  "Coexp TM1/TM5" is in this section (line 106-108).

line 115: judging from fig. 1B, the activity is not null, although significantly diminished compared to MS-DesKC. The same is true for lines 189-190: the activity is not lost completely, it is just diminished. Importantly, it is not clear whether this comes from lower expression level or lower activity of kinase per se.

It is true that the activity is not null, but it falls in background levels. In the new version we included the correction, lines 175,192, 196.

lines 231-237: it would be nice to support each of these statements with appropriate references.

This paragraph was replaced with a more appropriate one, which includes references (lines 323-329).

Round 2

Reviewer 2 Report

Authors addressed all points of review and improved the manuscript significantly. The topic is actual and interesting, and this paper will give a positive contribution to the field of signal transduction.